# Revisit Micro-batch Clipping: Adaptive Data Pruning via Gradient Manipulation

**Lun Wang**
Google
`lunwang@google.com`

## Abstract

Micro-batch clipping, a gradient clipping method, has recently shown potential in enhancing auto-speech recognition (ASR) model performance. However, the underlying mechanism behind this improvement remains mysterious, particularly the observation that only certain micro-batch sizes are beneficial. In this paper, we make the first attempt to explain this phenomenon. Inspired by recent data pruning research, we assume that specific training samples may impede model convergence during certain training phases. Under this assumption, the convergence analysis shows that micro-batch clipping can improve the convergence rate asymptotically at the cost of an additional constant bias that does not diminish with more training iterations. The bias is dependent on a few factors and can be minimized at specific micro-batch size, thereby elucidating the existence of the sweet-spot micro-batch size observed previously. We also verify the effectiveness of micro-batch clipping beyond speech models on vision and language models, and show promising performance gains in these domains. An exploration of potential limitations shows that micro-batch clipping is less effective when training data originates from multiple distinct domains.

## 1 Introduction

*Micro-batch clipping* (McMahan et al., 2018; Ponomareva et al., 2023) was initially introduced as a memory optimization technique for differentially private stochastic gradient descent (DP-SGD) (Abadi et al., 2016). The method groups per-example gradients within a mini-batch into smaller micro-batches, clips the average gradient of each micro-batch, and updates the model with the mean of the clipped micro-batch gradients to avoid materializing all per-example gradients in memory. Recent studies (Wang et al., 2024a;b) have surprisingly revealed that micro-batch clipping can enhance the performance of automatic speech recognition (ASR) models when used outside DP-SGD. This finding has sparked interest in exploring micro-batch clipping as a standalone optimization technique, offering potential benefits in model performance. However, the underlying mechanism driving this improvement remains poorly understood.

In this work, we aim to elucidate the behavior of micro-batch clipping through a combination of theoretical analysis and empirical evaluation. Specifically, we conceptualize micro-batch clipping as a specialized form of data pruning (Sorscher et al., 2022). Unlike traditional data pruning techniques, which deterministically exclude redundant data, micro-batch clipping adaptively suppresses samples that hinder convergence, referred to as "*draggers*", recognizing that a data sample's helpfulness can change throughout training. Guided by this intuition, we introduce Assumption 4.4 to capture certain properties of the draggers' gradients, which are later empirically verified in Section 5.1. Based on the assumption, we analyze the convergence-to-stationary-points rate for both standard SGD and micro-batch clipping on smooth loss manifolds and summarize the results in in Table 1. We can observe that the introduction of draggers slows down the convergence rate for SGD by a constant factor, while the usage of micro-batch clipping *asymptotically accelerates the convergence rate*, at the cost of an additional constant bias term. The bias term provides an explanation for the phenomenon observed in prior works (Wang et al., 2024a;b) where performance improvements are only seen with specific micro-batch sizes. Concretely, because this term does not vanish with increasing iterations, the performance gain is contingent upon the magnitude of this term being

Table 1: Convergence Rate

| | Dragger | Clipping | Convergence Rate |
|---|---|---|---|
| Thm 9 (Bu et al., 2024) | ✗ | ✗ | $\frac{1}{T^{1/4}}\sqrt{2L(\mathcal{L}_0 - \mathcal{L}_*) + \frac{\sigma_b^2}{B}}$ |
| Thm 4.1 | ✓ | ✗ | $\frac{1}{T^{1/4}}\sqrt{\frac{4L}{(1-\epsilon)^2}(\mathcal{L}_0 - \mathcal{L}_*) + \frac{2\sigma^2}{(1-\epsilon)B}}$ |
| Thm 4.2 | ✓ | ✓ | $\frac{\sigma}{(\sqrt{b}\epsilon - \sqrt{\epsilon(1-\epsilon)})\cdot(1+c)} + \frac{1}{\sqrt{T}}\cdot\mathcal{O}(\mathcal{L}_0 - \mathcal{L}_* + \frac{1}{2L})$ |

sufficiently small. According to Theorem 4.2, only certain micro-batch sizes minimize the bias term, thus explaining this effect.

To test the effectiveness of micro-batch clipping outside the speech domain, we apply it to vision and language models and observe promising performance improvements. Another interesting observation is that micro-batch clipping's efficacy diminishes when facing multi-domain training data, particularly when data sizes between domains are unbalanced. This may be attributed to the method hindering the convergence of domains with fewer samples by treating them as draggers.

## 2 BACKGROUND

### 2.1 MICRO-BATCH CLIPPING

While originally introduced in (McMahan et al., 2018) as a transition from record-level differential privacy to user-level differential privacy in federated learning, micro-batch clipping's widespread adoption in differential privacy libraries stems primarily from its memory efficiency (Ponomareva et al., 2023; Wang et al., 2024a;b). By requiring fewer gradients to be materialized, it reduces memory consumption, especially for large models, at the expense of a lower signal-to-noise ratio under the same differential privacy budget. Notably, per-core clipping (where the micro-batch size equals the per-core batch size) achieves complete memory parity with non-private training when data parallelism is employed.

The potential of micro-batch clipping as an optimization technique to improve model performance emerged from observations in (Wang et al., 2024a;b), where it's shown to reduce word error rate (WER) when training Conformer-based ASR models (Gulati et al., 2020). However, whether this benefit generalizes beyond specific model architectures and datasets remains an open question that this work seeks to address.

### 2.2 DATA PRUNING & ACTIVE LEARNING

A growing body of research (Toneva et al., 2018; Paul et al., 2021; Sorscher et al., 2022) highlights the detrimental effect of unhelpful or even harmful training data on model performance, particularly in large-scale datasets. Sorscher *et al.* (Sorscher et al., 2022) propose a metric-based approach to prune such data and alleviate inefficient power law scaling in theory. Although the approach can effectively reduce the size of the data to store, the method overlooks the dynamic nature of data utility, which can evolve throughout the training process.

Active learning (Bordes et al., 2005; Settles, 2009; Sener & Savarese, 2017; Birodkar et al., 2019; Mirzasoleiman et al., 2020; Emam et al., 2021; Karamcheti et al., 2021) offers an alternative approach to address this challenge by actively selecting the most beneficial training samples. Our work shares similarities with active learning in that data importance is determined dynamically during training based on gradient information. However, our approach is more fine-grained, implicitly adjusting data influence through adaptive gradient clipping rather than explicit sample selection.

## 3 METHODOLOGY: ADAPTIVE MICRO-BATCH CLIPPING

This section formally defines micro-batch clipping, focusing on the adaptive variant proposed in (Wang et al., 2024b), where the minimal L2 norm of all average micro-batch gradients is used

as the clipping bound, thus ensuring that all micro-batch gradients are clipped. We choose this variant because it avoids introducing clipping bound as an extra hyper-parameter, which requires further tuning, and (Wang et al., 2024b) shows that it performs comparably to clipping with manually selected clipping bound. For brevity, subsequent references to "micro-batch clipping" will refer to this adaptive variant.

Let $\mathcal{D}$ be the training set and we want to run stochastic gradient descent with adaptive micro-batch clipping. The mini-batch size is $B$ and learning rate is $\eta$. In each iteration, a mini-batch of training examples are picked and sharded into micro-batches of size $b$. The average gradient on each micro-batch will be calculated and clipped using the minimum L2 norm within all the micro-batch gradients. Then the clipped micro-batch gradients are summed to get the mini-batch gradient which is then used to update the model (or passed to the optimizer if adaptive optimizers are used.). The process is formalized in Algorithm 1.

---

**Algorithm 1** Pseudocode for SGD with adaptive micro-batch clipping. $\boldsymbol{g}_t$ represents mini-batch gradients in the $t^{th}$ iteration. $\hat{\boldsymbol{g}}_t^j$ represents the $j^{th}$ micro-batch's gradient in the $t^{th}$ iteration.

**Input: initial parameters $w_0$, loss function $\mathcal{L}$, training data $\mathcal{D}$, #iterations $T$, micro-batch size $b$, #mini-batch size $B$, learning rate $\eta$.**

1: **for** $t = 1, 2, \ldots T$ **do**
2:     $\{d_i^t\}_{i \in \{1,\ldots,B\}} \leftarrow \mathcal{D}$                               ▷ sample a mini-batch
3:     **for** $j = 1, 2, \ldots B/b$ **(parallelly) do**
4:         Load a micro-batch $\{d_i^t\}_{i \in \{b(j-1)+1,\ldots,bj\}}$
5:         $\hat{\boldsymbol{g}}_t^j = \nabla \mathcal{L}(\boldsymbol{w}_{t-1}, \{d_i^t\}_{i \in \{b(j-1)+1,\ldots,bj\}})$     ▷ obtain average gradient of a micro-batch
6:     $\rho_t = \min_j \|\hat{\boldsymbol{g}}_t^j\|_2$
7:     **for** $j = 1, 2, \ldots B/b$ **(parallelly) do**
8:         $\hat{\boldsymbol{g}}_t^j = \frac{\rho_t}{\|\hat{\boldsymbol{g}}_t^j\|_2} \cdot \hat{\boldsymbol{g}}_t^j$                       ▷ adaptive clipping
9:     $\boldsymbol{g}_t = \sum_{j \in \{1,\ldots,B/b\}} \hat{\boldsymbol{g}}_t^j$
10:    $\boldsymbol{w}_t = \boldsymbol{w}_{t-1} - \eta \cdot \boldsymbol{g}_t$                         ▷ update the model parameters

---

## 4   CONVERGENCE ANALYSIS

In this section, we provide the convergence-to-stationary-points analysis for SGD micro-batch clipping following (Bu et al., 2024) and show that it achieves asymptotically faster convergence at the cost of a constant bias under certain assumptions.

### 4.1   ASSUMPTIONS

Our analysis is based on a few assumptions, as listed below. Among these, Assumptions 4.1 and 4.2 are standard assumptions adopted from previous work (Chen et al., 2020; Bu et al., 2024). Assumption 4.3 is based on prior research (Chen et al., 2020; Bu et al., 2024), with a slight modification to accommodate Assumption 4.4, a new assumption that we introduce in this work to capture the existence of dragger examples.

**Assumption 4.1** (Lower bound of loss (Bu et al., 2024)). $\forall \boldsymbol{w}, \exists$ constant $\mathcal{L}_*$, s.t. $\mathcal{L}(\boldsymbol{w}) \geq \mathcal{L}_*$.

**Assumption 4.2** (Smoothness (Bu et al., 2024)). Define $\boldsymbol{g}(\boldsymbol{w}) := \frac{\partial \mathcal{L}(\boldsymbol{w})}{\partial \boldsymbol{w}}$. Then $\forall \boldsymbol{w}, \boldsymbol{v}, \exists$ constant $L \geq 0$ s.t.

$$\mathcal{L}(\boldsymbol{v}) - [\mathcal{L}(\boldsymbol{w}) + \boldsymbol{g}(\boldsymbol{w})^\top (\boldsymbol{v} - \boldsymbol{w})] \leq \frac{L}{2} \|\boldsymbol{w} - \boldsymbol{v}\|^2. \tag{1}$$

**Assumption 4.3** (Gradient distribution (Bu et al., 2024)). *With probability $1 - \epsilon$, the per-example gradient $\tilde{\boldsymbol{g}}$ is an i.i.d. symmetric unbiased estimator of $\boldsymbol{g}$ with bounded variance $\sigma^2$(Bu et al., 2024).*

$$w.p. \ 1 - \epsilon, \mathbb{E}[\tilde{\boldsymbol{g}}] = \boldsymbol{g}, \mathbb{E}[\|\tilde{\boldsymbol{g}} - \boldsymbol{g}\|^2] \leq \sigma^2$$

*With probability $\epsilon$, the per-example gradient is a dragger*

$$w.p. \ \epsilon, \tilde{\boldsymbol{g}} = \boldsymbol{\mu}$$

Assumption 4.4 is introduced to formalize the observation that certain gradients within the training data may impede the model's convergence. First, we assume that these "dragger" gradients are orthogonal to the gradients of benign examples, following the intuition that the cosine similarity between data from different domains should be low. The intuition is both motivated and supported by empirical evidence presented in Section 5.1. Second, we assume the L2 norm of the dragger gradient is both lower- and upper-bounded by the L2 norm of the expected benign gradient scaled by a constant factor. The lower-bound captures the intuition that dragger gradients should be large enough to slow down the convergence. Empirical evidence for the intuition is presented in Section 5.3. The upper bound captures the fact that these detrimental gradients are not the result of adversarial manipulation but arise naturally within the vast and diverse training dataset.

**Assumption 4.4** (Dragger). *The dragger gradients are orthogonal to the benign gradient subspace. Furthermore, we assume that the ratio of the dragger's norm to the expected benign gradient norm is bounded both above and below.*

$$\boldsymbol{\mu} \perp \tilde{\boldsymbol{g}}, c\|\boldsymbol{g}\| \leq \|\boldsymbol{\mu}\| \leq C\|\boldsymbol{g}\|$$

## 4.2 CONVERGENCE RATE OF STANDARD SGD WITH DRAGGER GRADIENTS

We first provide the convergence to stationary point of standard SGD result under the assumptions above.

**Theorem 4.1.** *Under Assumption 4.1, 4.2, 4.3 (without the symmetry assumption), 4.4, running SGD for $T$ iterations gives, for $\eta = \frac{1}{L\sqrt{T}}$,*

$$\min_t \mathbb{E}\|\boldsymbol{g}_t\| \leq \frac{1}{T^{1/4}} \sqrt{\frac{4L}{(1-\epsilon)^2}(\mathcal{L}_0 - \mathcal{L}_*) + \frac{2\sigma^2}{(1-\epsilon)B}}$$

*Proof.* In the standard SGD,

$$\boldsymbol{w}_{t+1} = \boldsymbol{w}_t - \frac{\eta}{B}\sum \tilde{\boldsymbol{g}}_{t,i}$$

where $\tilde{\boldsymbol{g}}_{t,i}$ here represents $i^{th}$ per-example gradient sampled from the distribution described by Assumption 4.3 and 4.4.

By Assumption 4.2,

$$\mathcal{L}_{t+1} - \mathcal{L}_t \leq \boldsymbol{g}_t^\top(\boldsymbol{w}_{t+1} - \boldsymbol{w}_t) + \frac{L}{2}\|\boldsymbol{w}_{t+1} - \boldsymbol{w}_t\|^2$$

$$= -\frac{\eta\boldsymbol{g}_t^\top}{B}\Big(\underbrace{\sum \tilde{\boldsymbol{g}}_{t,i}}_{\text{benign}} + \underbrace{\sum \tilde{\boldsymbol{g}}_{t,i}}_{\text{dragger}}\Big) + \frac{L\eta^2}{2B^2}\|\underbrace{\sum \tilde{\boldsymbol{g}}_{t,i}}_{\text{benign}} + \underbrace{\sum \tilde{\boldsymbol{g}}_{t,i}}_{\text{dragger}}\|^2$$

The expected improvement at one iteration is

$$\mathbb{E}[\mathcal{L}_{t+1} - \mathcal{L}_t|\boldsymbol{w}_t] \leq -(1-\epsilon)\eta\boldsymbol{g}_t^\top\mathbb{E}_{\text{benign}}[\tilde{\boldsymbol{g}}_{t,i}] - \epsilon\eta\boldsymbol{g}_t^\top\mathbb{E}_{\text{dragger}}[\tilde{\boldsymbol{g}}_{t,i}] + \frac{L\eta^2}{2B^2}\mathbb{E}\|\underbrace{\sum \tilde{\boldsymbol{g}}_{t,i}}_{\text{benign}} + \underbrace{\sum \tilde{\boldsymbol{g}}_{t,i}}_{\text{dragger}}\|^2$$

$$= -(1-\epsilon)\eta\|\boldsymbol{g}_t\|^2 + \frac{L\eta^2}{2B^2}\mathbb{E}\|\underbrace{\sum \tilde{\boldsymbol{g}}_{t,i}}_{\text{benign}}\|^2 + \frac{L\eta^2}{2B^2}\mathbb{E}\|\underbrace{\sum \tilde{\boldsymbol{g}}_{t,i}}_{\text{dragger}}\|^2$$

$$\leq -(1-\epsilon)\eta\|\boldsymbol{g}_t\|^2 + \frac{(1-\epsilon)^2L\eta^2}{2}(\|\boldsymbol{g}_t\|^2 + \frac{\sigma^2}{(1-\epsilon)B}) + \frac{\epsilon^2L\eta^2}{2}\|\boldsymbol{\mu}_t\|^2$$

$$\leq \Big(\frac{(1-\epsilon)^2L\eta^2}{2} - (1-\epsilon)\eta + \frac{\epsilon^2L\eta^2C^2}{2}\Big)\|\boldsymbol{g}_t\|^2 + \frac{(1-\epsilon)L\eta^2\sigma^2}{2B}$$

The second equation follows from Assumption 4.4 that the benign gradients are perpendicular to the dragger gradients. The fourth inequality is based on Assumption 4.4 that the dragger gradient norm

is upper bounded. To make sure the coefficient is negative, we require that

$$\frac{(1-\epsilon)^2 L\eta^2}{2} - (1-\epsilon)\eta + \frac{\epsilon^2 L\eta^2 C^2}{2} < 0 \Rightarrow C < \sqrt{\frac{2(1-\epsilon)\eta - (1-\epsilon)^2 L\eta^2}{\epsilon^2 L\eta^2}}$$

For arithmetic convenience, we choose $C = \sqrt{\frac{2(1-\epsilon)\eta - (1-\epsilon)^2 L\eta^2}{2\epsilon^2 L\eta^2}}$. Now we do a telescoping sum over the iterations

$$\mathcal{L}_0 - \mathcal{L}_* \geq \mathcal{L}_0 - \mathbb{E}\mathcal{L}_T = \sum_t \mathbb{E}(\mathcal{L}_t - \mathcal{L}_{t+1})$$

$$\geq \left(\frac{(1-\epsilon)\eta}{2} - \frac{(1-\epsilon)^2 L\eta^2}{4}\right)\mathbb{E}\left(\sum_t \|\boldsymbol{g}_t\|^2\right) - \frac{(1-\epsilon)L\eta^2\sigma^2 T}{2B}$$

We apply the same learning rate as in (Bu et al., 2024) $\eta = \frac{1}{L\sqrt{T}}$.

$$\mathcal{L}_0 - \mathcal{L}_* \geq \left(\frac{1-\epsilon}{2L\sqrt{T}} - \frac{(1-\epsilon)^2}{4LT}\right)\mathbb{E}\left(\sum_t \|\boldsymbol{g}_t\|^2\right) - \frac{(1-\epsilon)\sigma^2}{2LB}$$

$$\geq \frac{(1-\epsilon)^2}{4L\sqrt{T}}\mathbb{E}\left(\sum_t \|\boldsymbol{g}_t\|^2\right) - \frac{(1-\epsilon)\sigma^2}{2LB}$$

and finally

$$\min_t \mathbb{E}\|\boldsymbol{g}_t\|^2 \leq \frac{1}{\sqrt{T}}\left(\frac{4L}{(1-\epsilon)^2}(\mathcal{L}_0 - \mathcal{L}_*) + \frac{2\sigma^2}{(1-\epsilon)B}\right)$$

Using Jensen's inequality, we can have

$$\min_t \mathbb{E}\|\boldsymbol{g}_t\| \leq \frac{1}{T^{1/4}}\sqrt{\frac{4L}{(1-\epsilon)^2}(\mathcal{L}_0 - \mathcal{L}_*) + \frac{2\sigma^2}{(1-\epsilon)B}}$$

$\square$

In comparison to Theorem 9 of (Bu et al., 2024), SGD with draggers demonstrates the same asymptotic convergence rate, albeit with a larger constant coefficient. This observation aligns with our intuition that the presence of dragger gradients can hinder the model's convergence speed.

### 4.3 CONVERGENCE RATE OF MICRO-BATCH CLIPPED SGD WITH DRAGGER GRADIENTS

In this subsection, we present the convergence analysis for micro-batch clipping. To simplify the algebra, we assume a uniform lower bound for the clipping bound ($\rho_t$) used in each iteration. Note that this assumption doesn't impact the asymptotic behavior. For conciseness, the proofs of the supporting lemmas are provided in the appendix.

**Theorem 4.2.** *Under Assumption 4.1, 4.2, 4.3, 4.4, running micro-batch clipped SGD for $T$ iterations gives, for $\eta = \frac{1}{L\sqrt{T}}$, micro-batch size $b$ and dragger probability $\epsilon \geq \frac{1}{b+1}$,*

$$\min_t \mathbb{E}\|\boldsymbol{g}_t\| \leq \frac{\sigma}{(\sqrt{b\epsilon} - \sqrt{\epsilon(1-\epsilon)}) \cdot (1+c)}$$

$$+ \frac{1}{\sqrt{T}} \cdot \frac{\sqrt{b\epsilon}}{\sqrt{b\epsilon} - \sqrt{1-\epsilon}} \cdot \frac{2L(1+2\epsilon C)}{1-\epsilon} \cdot (\mathcal{L}_0 - \mathcal{L}_* + \frac{1}{2L})$$

*Proof Sketch.* In the micro-batch clipped SGD with micro-batch size $b$, the update rule is as follow:

$$\boldsymbol{w}_{t+1} = \boldsymbol{w}_t - \frac{\eta b}{B} \sum_i \frac{\hat{\boldsymbol{g}}_{t,i}}{\|\hat{\boldsymbol{g}}_{t,i}\|} \tag{2}$$

Note that here $\hat{g}_{t,i}$ represents $i^{th}$ micro-batch gradient in the $t^{th}$ iteration. For notation simplicity, we omit the subscripts if it's clear from context.

By plugging in the update from Equation 2 in the smoothness assumption 4.2 and take conditional expectation, the expected improvement at one iteration is

$$\mathbb{E}[\mathcal{L}_{t+1} - \mathcal{L}_t | \boldsymbol{w}_t] \leq -\eta \boldsymbol{g}_t^\top \mathbb{E}\frac{\hat{\boldsymbol{g}}_t}{\|\hat{\boldsymbol{g}}_t\|} + \frac{L\eta^2}{2}$$

Re-write $\hat{\boldsymbol{g}}_t = (1 - \epsilon)\boldsymbol{g}_t + \epsilon\boldsymbol{\mu}_t + \Delta_t$, where $\Delta_t$ is a zero-centered random variable whose expected L2 norm is bounded as below. The proof can be found in Appendix A.

**Lemma 4.3.** *The expected L2 norm of $\Delta_t$ is upper-bounded by.*

$$\mathbb{E}\|\Delta_t\| \leq \sqrt{\frac{\epsilon(1 - \epsilon)\|\boldsymbol{g}_t\|^2 + \epsilon(1 - \epsilon)\|\boldsymbol{\mu}_t\|^2 + (1 - \epsilon)\sigma^2}{b}}$$

.

To lower bound $\boldsymbol{g}_t^\top \mathbb{E}\frac{\hat{\boldsymbol{g}}_t}{\|\hat{\boldsymbol{g}}_t\|}$, we follow (Bu et al., 2024) to use the hyperplane perpendicular to $\boldsymbol{g}_t$ to divide the support of $\Delta_t$ into two half-spaces where denote the positive half as: $H_+ := \{v : \boldsymbol{g}^\top v > 0\}$. Then using the symmetry assumption 4.3, we have

$$\begin{aligned}
\boldsymbol{g}_t^\top \mathbb{E}\frac{\hat{\boldsymbol{g}}_t}{\|\hat{\boldsymbol{g}}_t\|} &= \mathbb{E}\Big(\frac{(1 - \epsilon)\|\boldsymbol{g}_t\|^2 + \boldsymbol{g}_t^\top \Delta_t}{\|(1 - \epsilon)\boldsymbol{g}_t + \epsilon\boldsymbol{\mu}_t + \Delta_t\|}\Big) \\
&= \frac{1}{2}\mathbb{E}\Big(\frac{(1 - \epsilon)\|\boldsymbol{g}_t\|^2 + \boldsymbol{g}_t^\top \Delta_t}{\|(1 - \epsilon)\boldsymbol{g}_t + \epsilon\boldsymbol{\mu}_t + \Delta_t\|}\Big|\Delta_t \in H_+\Big) + \frac{1}{2}\mathbb{E}\Big(\frac{(1 - \epsilon)\|\boldsymbol{g}_t\|^2 - \boldsymbol{g}_t^\top \Delta_t}{\|(1 - \epsilon)\boldsymbol{g}_t + \epsilon\boldsymbol{\mu}_t - \Delta_t\|}\Big|\Delta_t \in H_+\Big) \\
&= \frac{1}{2}\mathbb{E}\Big(\underbrace{\frac{(1 - \epsilon)\|\boldsymbol{g}_t\|^2 + \boldsymbol{g}_t^\top \Delta_t}{\|(1 - \epsilon)\boldsymbol{g}_t + \epsilon\boldsymbol{\mu}_t + \Delta_t\|} + \frac{(1 - \epsilon)\|\boldsymbol{g}_t\|^2 - \boldsymbol{g}_t^\top \Delta_t}{\|(1 - \epsilon)\boldsymbol{g}_t + \epsilon\boldsymbol{\mu}_t - \Delta_t\|}}_{\star}\Big|\Delta_t \in H_+\Big) \\
&= \frac{1}{2}\mathbb{E}\big(\star\big|\Delta_t \in H_+, \|\Delta_t\| \leq \epsilon\|\boldsymbol{\mu}_t\| + \epsilon\|\boldsymbol{g}_t\|\big)\mathbb{P}\big(\|\Delta\| \leq \epsilon\|\boldsymbol{\mu}_t\| + \epsilon\|\boldsymbol{g}_t\|\big) \\
&\quad + \frac{1}{2}\mathbb{E}\big(\star\big|\Delta_t \in H_+, \|\Delta_t\| > \epsilon\|\boldsymbol{\mu}_t\| + \|\boldsymbol{g}_t\|\big)\mathbb{P}\big(\|\Delta_t\| > \epsilon\|\boldsymbol{\mu}_t\| + \epsilon\|\boldsymbol{g}_t\|\big)
\end{aligned}$$

Lemma 4.4 tells us that $\star$ is always non-negative under Assumption 4.4, and thus we can only keep the first term. The proof is provided in Appendix B.

**Lemma 4.4.** *If $\Delta_t \in H_+$ and $\boldsymbol{\mu}_t \perp \boldsymbol{g}_t, \boldsymbol{\mu}_t \perp \Delta_t$, then $\frac{(1-\epsilon)\|\boldsymbol{g}_t\|^2 + \boldsymbol{g}_t^\top \Delta_t}{\|(1-\epsilon)\boldsymbol{g}_t + \epsilon\boldsymbol{\mu}_t + \Delta_t\|} + \frac{(1-\epsilon)\|\boldsymbol{g}_t\|^2 - \boldsymbol{g}_t^\top \Delta_t}{\|(1-\epsilon)\boldsymbol{g}_t + \epsilon\boldsymbol{\mu}_t - \Delta_t\|} \geq 0$.*

$$\begin{aligned}
\boldsymbol{g}_t^\top \mathbb{E}\frac{\hat{\boldsymbol{g}}_t}{\|\hat{\boldsymbol{g}}_t\|} &\geq \frac{1}{2}\mathbb{E}\big(\star\big|\Delta_t \in H_+, \|\Delta_t\| \leq \epsilon\|\boldsymbol{\mu}_t\| + \epsilon\|\boldsymbol{g}_t\|\big)\mathbb{P}\big(\|\Delta_t\| \leq \epsilon\|\boldsymbol{\mu}_t\| + \epsilon\|\boldsymbol{g}_t\|\big) \\
&\geq \frac{1}{2}\mathbb{E}\Big(\frac{(1 - \epsilon)\|\boldsymbol{g}_t\|^2}{(1 - \epsilon)\|\boldsymbol{g}_t\| + \epsilon\|\boldsymbol{\mu}_t\| + \|\Delta_t\|}\Big|\Delta_t \in H_+, \|\Delta_t\| \leq \epsilon\|\boldsymbol{\mu}_t\| + \epsilon\|\boldsymbol{g}_t\|\Big)\Big(1 - \frac{\mathbb{E}\|\Delta_t\|}{\epsilon\|\boldsymbol{\mu}_t\| + \epsilon\|\boldsymbol{g}_t\|}\Big) \\
&\geq \frac{1}{2}\cdot\frac{(1 - \epsilon)\|\boldsymbol{g}_t\|^2}{\|\boldsymbol{g}_t\| + 2\epsilon\|\boldsymbol{\mu}_t\|}\cdot\Big(1 - \frac{1}{\sqrt{b}}\cdot\frac{\sqrt{\epsilon(1 - \epsilon)\|\boldsymbol{g}_t\|^2 + \epsilon(1 - \epsilon)\|\boldsymbol{\mu}_t\|^2 + (1 - \epsilon)\sigma^2}}{\epsilon\|\boldsymbol{\mu}_t\| + \epsilon\|\boldsymbol{g}_t\|}\Big) \\
&\geq \frac{1 - \epsilon}{2(1 + 2\epsilon C)}\cdot\Big(\Big(1 - \sqrt{\frac{1 - \epsilon}{b\epsilon}}\Big)\|\boldsymbol{g}_t\| - \frac{1}{\sqrt{b}}\cdot\frac{\sigma}{\epsilon(1 + c)}\Big)
\end{aligned}$$

, where the second inequality follows from Markov's inequality.

Thus we have

$$\mathbb{E}[\mathcal{L}_{t+1} - \mathcal{L}_t | \boldsymbol{w}_t] \leq -\eta\cdot\frac{1 - \epsilon}{2(1 + 2\epsilon C)}\cdot\Big(\Big(1 - \sqrt{\frac{1 - \epsilon}{b\epsilon}}\Big)\|\boldsymbol{g}_t\| - \frac{1}{\sqrt{b}}\cdot\frac{\sigma}{\epsilon(1 + c)}\Big) + \frac{L\eta^2}{2}$$

Now we do a telescoping sum over the iterations

$$\mathcal{L}_0 - \mathcal{L}_* \geq \mathcal{L}_0 - \mathbb{E}\mathcal{L}_T = \sum_t \mathbb{E}(\mathcal{L}_t - \mathcal{L}_{t+1})$$

$$\geq \eta \cdot \frac{1-\epsilon}{2(1+2\epsilon C)} \cdot \left((1 - \sqrt{\frac{1-\epsilon}{b\epsilon}})\|\boldsymbol{g}_t\| - \frac{1}{\sqrt{b}} \cdot \frac{\sigma}{\epsilon(1+c)}\right) - \frac{L\eta^2 T}{2}$$

We apply the same learning rate as in (Bu et al., 2024) $\eta = \frac{1}{L\sqrt{T}}$.

$$\mathcal{L}_0 - \mathcal{L}_* \geq \frac{1-\epsilon}{2L\sqrt{T}(1+2\epsilon C)} \cdot \left((1 - \sqrt{\frac{1-\epsilon}{b\epsilon}})\|\boldsymbol{g}_t\| - \frac{1}{\sqrt{b}} \cdot \frac{\sigma}{\epsilon(1+c)}\right) - \frac{1}{2L}$$

and finally

$$\min_t \mathbb{E}\|\boldsymbol{g}_t\| \leq \min_t \frac{1}{T}\mathbb{E}\sum_t \|\boldsymbol{g}_t\| \leq \frac{\sigma}{(\sqrt{b\epsilon} - \sqrt{\epsilon(1-\epsilon)}) \cdot (1+c)}$$

$$+ \frac{1}{\sqrt{T}} \cdot \frac{\sqrt{b\epsilon}}{\sqrt{b\epsilon} - \sqrt{1-\epsilon}} \cdot \frac{2L(1+2\epsilon C)}{1-\epsilon} \cdot (\mathcal{L}_0 - \mathcal{L}_* + \frac{1}{2L})$$

$$\square$$

## 5 EMPIRICAL EVALUATION

In this section, we'd like to answer the following 3 questions: 1) Does Assumption 4.4 hold in practice? 2) Does the hypothesis on $c$ to explain the sweet spot of micro-batch size hold in practice? 3) Does micro-batch clipping help improve performance beyond speech tasks?

### 5.1 EMPIRICAL EVIDENCE FOR ASSUMPTION 4.4

**Experiment Setup.** To answer question 1), we adopt the experimental setup from Wang *et al.* (Wang et al., 2024a). Specifically, we fine-tune 600M Conformer XL (Zhang et al., 2020) models on the LibriSpeech dataset (Panayotov et al., 2015) and handcrafted canaries (Wang et al., 2024b). The model's encoder is pre-trained using BEST-RQ (Chiu et al., 2022) on the LibriLight dataset (Kahn et al., 2020). The key advantage of this setup is the ability to treat the inserted canaries as surrogate draggers, thereby circumventing the technical challenge of identifying natural draggers in large models.

We compute the cosine similarity between 100 randomly selected pairs of dragger and benign gradients. For calibration purposes, we also compute the cosine similarity between 100 pairs of benign gradients.

**Evaluation Results.** The results are summarized in Figure 1. We observe that the magnitude of the cosine similarity between dragger and benign gradients is significantly smaller than the cosine similarity within benign gradients, with a negligible standard deviation. This observation supports Assumption 4.4 that dragger gradients are orthogonal to the benign gradient subspace.

### 5.2 HOW $c$ CHANGES WITH MICRO-BATCH SIZE?

**Experiment Setup.** Again we re-use the setup in (Wang et al., 2024a). First, we run micro-batch clipping with micro-batch size 1, 4 (per-core batch size), 512 (mini-batch size) to substantiate the existence of an optimal micro-batch size that yields maximum performance. Performance is assessed using word error rate (WER) on two splits of the LibriSpeech test dataset: test-clean, consisting of relatively clean utterances, and test-other, consisting of noisier utterances.

In the second step, the following proxy estimation is used to monitor the fluctuations in the value of $c$.

$$\hat{c} := \frac{\text{mean}(\|\text{benign gradients}\|)}{\text{mean}(\|\text{dragger gradients}\|)}$$

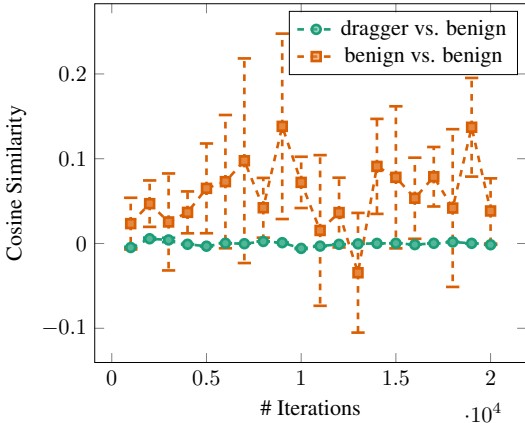

Figure 1: Cosine Similarity between Crafted Dragger and Benign Examples.

To estimate the numerator, we use the mean norm of 100 LibriSpeech samples after trimming the largest 10% to avoid the effect of hard-to-track natural draggers. To estimate the denominator, we use the mean of the gradient norms of 100 inserted canaries. The proxy $\hat{c}$ is computed for both non-private training and micro-batch clipping scenarios with micro-batch sizes 1, 4 and 512.

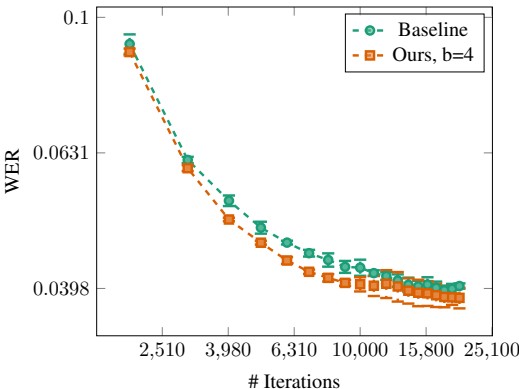

Figure 2: Convergence curve for LibriSpeech experiments. Both x-axis and y-axis are log-scale to highlight the convergence speed advantage.

**Evaluation Results.** First, in Table 2, we can observe that when micro-batch size is 4, the model achieves the best performance across all settings, improving test-other WER by 4.8% relatively and test-clean by 4.1% relatively. Furthermore, we note that micro-batch clipping does not enhance performance when the micro-batch size is either 1 or 512. Instead, these configurations degrade WER on both the test-other and test-clean splits. As per Theorem 4.2, this degradation is attributed to the presence of a large constant term when the micro-batch size is not optimally selected. Furthermore, we note that micro-batch clipping does not enhance performance when the micro-batch size is either 1 or 512. Instead, these configurations degrade WER on both the test-other and test-clean splits. As per Theorem 4.2, this degradation is attributed to the presence of a large constant term when the micro-batch size is not optimally selected. Besides, from Figure 2, we can tell see the convergence rate advantage as predicted.

Second, the values of $\hat{c}$ for different micro-batch sizes are shown in Table 3. The results conform with the conjecture that $\hat{c}$ is decreasing with respect to the micro-batch size $b$. This observation provides a strong evidence to support the explanation for the existence of the optimal micro-batch size.

*Remark.* We'd like to note that the difference in proxy $\hat{c}$ is still not big enough to level out the influence of $b$ in the constant term in Theorem 4.2, and this might be attributed to the several factors including the convergence rate bound not strictly tight, the noise in $\hat{c}$ measurement, and systematic

Table 2: Best WER on LibriSpeech within 20K fine-tuning steps. Mean and standard deviation are calculated across 3 runs. **Bold** highlights best. The checkpoint is picked to optimize test-other split.

| Test set | Baseline | Ours, $b = 1$ | Ours, $b = 4$ | Ours, $b = 512$ |
|---|---|---|---|---|
| test-clean | $1.93 \pm 0.04$ | $2.40 \pm 0.13$ | $\mathbf{1.85} \pm 0.07$ | $1.94 \pm 0.00$ |
| test-other | $3.99 \pm 0.01$ | $4.86 \pm 0.24$ | $\mathbf{3.80} \pm 0.05$ | $4.03 \pm 0.03$ |

bias between $\hat{c}$ and $c$. However, we still believe that the trend of $\hat{c}$ is promising as an evidence for our explanation for sweet-spot micro-batch sizes, and we deem getting a stricter convergence bound and more accurate relationship between $b$ and $c$ important future directions to guide the choice of micro-batch sizes.

Table 3: The values of $\hat{c}$ under different micro-batch sizes.

| Micro-batch size | $b = 1$ | $b = 4$ | $b = 512$ |
|---|---|---|---|
| $\hat{c}$ | 6.18 | 5.74 | 4.42 |

### 5.3 IS MICRO-BATCH CLIPPING EFFECTIVE FOR MODELS BEYOND SPEECH?

To address question 3), we conduct an evaluation of micro-batch clipping on vision models and language models. In accordance with the findings of (Wang et al., 2024b), we default to a micro-batch size equivalent to the per-core batch size due to its memory efficiency. It is important to note that alternative micro-batch sizes may necessitate careful adjustment to mitigate potential memory overflow issues.

#### 5.3.1 VISION TASKS

**Experiment Setup:** For vision tasks, we choose apply micro-batch clipping to the DeiT-B model proposed in (Touvron et al., 2021). We train DeiT-B from scratch on the ImageNet dataset (Deng et al., 2009) and achieves parity with the reported Top-1 accuracy in (Touvron et al., 2021). The mini-batch size used is 4096 and the micro-batch size is 32, the same as the per-core batch size for computational efficiency.

**Evaluation Results:** The results are summarized in Table 4. Adding micro-batch clipping improves the Top-1 accuracy by 1.5% and Top-5 accuracy by 1.0%.

Table 4: ViT trained w/ or w/o adaptive micro-batch clipping.

| | Top-1 accuracy (%) | Top-5 accuracy (%) |
|---|---|---|
| Baseline | 81.8 | 95.8 |
| Ours | **83.3** | **96.8** |

#### 5.3.2 LANGUAGE TASKS

**Experiment Setup:** For language tasks, we fine-tune a T5 model (Raffel et al., 2020) on the superGlue dataset (Wang et al., 2019). The model is pre-trained on the C4 dataset (Raffel et al., 2020) for 1M steps. The mini-batch size is 2048, and the mini-batch size is 16, the same as the per-core batch size. Note that the test set for superGlue is private so we report the performance on the validation set. Since we do not use any information of the validation set to improve the micro-batch clipping part, the improvement shown is fair. Also we use accuracy across all subsets of superGlue to be able to compute the weighted average accuracy as a unified metric to compare 2 models.

**Evaluation Results:** Table 5 summarizes our findings. While micro-batch clipping only marginally improves average accuracy (0.1%), it significantly boosts performance on the largest, unbalanced subset, ReCoRD (0.4%), while negatively impacting smaller datasets. This suggests varying gradient distributions across subsets, with micro-batch clipping suppressing gradients from smaller subsets,

which conforms with the observation that ReCoRD has lower accuracy when trained with other data mixed (83.9%) than when trained alone (84.5%). Training solely on ReCoRD yields a 0.3% improvement as shown in Table 6, reinforcing this hypothesis.

Table 5: T5 trained w/ or w/o adaptive micro-batch clipping.

| Accuracy (%) | wt avg | BoolQ | CB | COPA | MultiRC | ReCoRD | RTE | WiC | WSC |
|---|---|---|---|---|---|---|---|---|---|
| ‖Train‖ | - | 9427 | 250 | 400 | 5100 | 101k | 2500 | 6000 | 554 |
| Baseline | 85.9 | 93.4 | **97.8** | **88.9** | **93.4** | 83.9 | **97.6** | **89.5** | **70.3** |
| Ours | **86.0** | 91.8 | 95.6 | 88.3 | 91.6 | **84.3** | 95.7 | 88.5 | **70.3** |

These results highlight a crucial lesson: micro-batch clipping, while beneficial when data comes from the same domain but is noisy, can be detrimental with unbalanced multi-domain data. In such scenarios, micro-batch clipping may hinder certain domains to favor others and should be avoided.

Table 6: T5 trained w/ or w/o adaptive micro-batch clipping.

| Accuracy (%) | Baseline | Ours |
|---|---|---|
| ReCoRD | 84.5 | **84.8** |

## 6 CONCLUSION & DISCUSSION

In this paper, we revisit micro-batch clipping, originally proposed in the context of differential privacy, from the lens of data pruning, inspired by recent observations made by (Wang et al., 2024a;b).

Our convergence analysis demonstrates that micro-batch clipping can asymptotically accelerate the convergence rate for smooth loss functions. To elucidate the optimal micro-batch size falling between 1 and the mini-batch size, we introduce the concept of "dragger gradients." Combining this concept with our convergence analysis reveals a constant term minimized at a value between 1 and the mini-batch size.

Our analysis and explanation hinge on two novel hypotheses, which we empirically verify. We also extend the application of micro-batch clipping to vision and language models. We find that when the input data is single-domain, micro-batch clipping can still enhance the performance of these models.

**Limitations.** Despite the demonstrated effectiveness of micro-batch clipping across speech, vision and language models, our research reveals limitations that warrant further investigation. As evidenced in Section 5.3.2, micro-batch clipping can adversely affect model performance when the input data originates from multiple distinct domains. This constraint limits the applicability of micro-batch clipping in scenarios where data diversity is prevalent.

Furthermore, this limitation may raise potential fairness concerns, akin to other data pruning methods pointed out by (Vysogorets et al., 2024). The preferential treatment of specific data domains could inadvertently introduce biases and inequities in model outcomes. These limitations underscore the need for a refined micro-batch clipping approach that can effectively handle multi-domain data.

**Future Work.** Several intriguing avenues for future research emerge from this paper. Firstly, the memory overhead introduced by micro-batch clipping remains a challenge. While data sharding with a micro-batch size equal to the per-core batch size (Wang et al., 2024a;b) can mitigate this issue, the growing size of modern models necessitates the development of alternative solutions to ensure the computational efficiency of micro-batch clipping in training scenarios with model sharding alone.

Secondly, from a theoretical perspective, a more rigorous relationship between micro-batch size $b$ and $c$ is needed, as it could guide the choice of the appropriate micro-batch size to optimize performance.

Finally, we are eager to explore a broader range of gradient-based data pruning methods. This includes investigating different variations of clipping, as well as other innovative gradient manipulation techniques based on diverse heuristics.

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

## A  PROOF FOR LEMMA 4.3

*Proof.* First, we upper-bound the variance for the per-example gradients following the distribution described in Assumption 4.3.

$$
\begin{aligned}
\mathbb{E}\|\tilde{\boldsymbol{g}} - \mathbb{E}[\tilde{\boldsymbol{g}}]\|^2 &= \mathbb{E}\|\tilde{\boldsymbol{g}}\|^2 - \|\mathbb{E}\tilde{\boldsymbol{g}}\|^2 \\
&\leq (1-\epsilon)\big(\|\boldsymbol{g}\|^2 + \sigma^2\big) + \epsilon\|\boldsymbol{\mu}\|^2 - \|(1-\epsilon)\boldsymbol{g} + \epsilon\boldsymbol{\mu}\|^2 \\
&= \epsilon(1-\epsilon)\|\boldsymbol{g}\|^2 + \epsilon(1-\epsilon)\|\boldsymbol{\mu}\|^2 + (1-\epsilon)\sigma^2
\end{aligned}
$$

Second, after the micro-batch averaging,

$$
\begin{aligned}
\mathbb{E}\|\hat{\boldsymbol{g}} - \mathbb{E}[\hat{\boldsymbol{g}}]\|^2 &= \frac{1}{b^2}\mathbb{E}\big(\|\sum(\tilde{\boldsymbol{g}} - \mathbb{E}\tilde{\boldsymbol{g}})\|^2\big) \\
&= \frac{1}{b^2}\sum\mathbb{E}\big(\|\tilde{\boldsymbol{g}} - \mathbb{E}\tilde{\boldsymbol{g}}\|^2\big) \\
&= \frac{\epsilon(1-\epsilon)\|\boldsymbol{g}\|^2 + \epsilon(1-\epsilon)\|\boldsymbol{\mu}\|^2 + (1-\epsilon)\sigma^2}{b}
\end{aligned}
$$

, where the second equation follows from the fact that per-example gradients are independent from each other. □

# B PROOF FOR LEMMA 4.4

*Proof.* If $\boldsymbol{g}_t^\top \Delta_t \leq (1-\epsilon)\|\boldsymbol{g}_t\|^2$, then the lemma obviously holds, so we only need to prove for the case when $\boldsymbol{g}_t^\top \Delta_t > (1-\epsilon)\|\boldsymbol{g}_t\|^2$.

$$\epsilon^2 \|\boldsymbol{g}_t\|^2 \|\boldsymbol{\mu}_t\| + \|\boldsymbol{g}_t\|^2 \|\Delta\|^2 \geq (\boldsymbol{g}_t^\top \Delta_t)^2$$

$$\Rightarrow \frac{((1-\epsilon)\|\boldsymbol{g}_t\|^2 + \boldsymbol{g}_t^\top \Delta_t)^2}{(1-\epsilon)^2 \|\boldsymbol{g}_t\|^2 + \epsilon^2 \|\boldsymbol{\mu}\|^2 + \|\Delta_t\|^2 + 2(1-\epsilon)\boldsymbol{g}_t^\top \Delta_t}$$

$$\geq \frac{((1-\epsilon)\|\boldsymbol{g}_t\|^2 - \boldsymbol{g}_t^\top \Delta_t)^2}{(1-\epsilon)^2 \|\boldsymbol{g}_t\|^2 + \epsilon^2 \|\boldsymbol{\mu}\|^2 + \|\Delta_t\|^2 - 2(1-\epsilon)\boldsymbol{g}_t^\top \Delta_t}$$

$$\Leftrightarrow \frac{(1-\epsilon)\|\boldsymbol{g}_t\|^2 + \boldsymbol{g}_t^\top \Delta_t}{\|(1-\epsilon)\boldsymbol{g}_t + \epsilon\boldsymbol{\mu}_t + \Delta_t\|} + \frac{(1-\epsilon)\|\boldsymbol{g}_t\|^2 - \boldsymbol{g}_t^\top \Delta_t}{\|(1-\epsilon)\boldsymbol{g}_t + \epsilon\boldsymbol{\mu}_t - \Delta_t\|} \geq 0$$

$\square$

