# OpenReview forum: "Revisit Micro-batch Clipping: Adaptive Data Pruning via Gradient Manipulation"
_ICLR.cc/2025/Conference — ICLR 2025 Poster_

### Official Review · Reviewer_G2Xp · 2024-10-17

**Soundness:** 3
**Presentation:** 2
**Contribution:** 2
**Rating:** 6
**Confidence:** 4

**Summary:**

This paper presents a theoretical analysis of why the micro-batch clipping method, which has been shown to improve performance in automatic speech recognition (ASR) tasks, is only effective for specific micro-batch sizes. The authors first categorize mini-batch samples into benign samples and dragger samples, followed by a convergence analysis. This analysis provides an explanation of the sweet spot for micro-batch size. Finally, the paper demonstrates that micro-batch clipping is also effective in both vision and language tasks.

**Strengths:**

- The topic of theoretically analyzing the batch-size dependency issue of micro-batch clipping in training ASR models, which has shown empirical success, is both meaningful and interesting.

- Despite the mathematical rigor throughout the paper, the overall writing is easy to read and follow; in particular, both the content and proof of Theorem 4.1 are easy to understand.

- In the case of the ASR task, the paper presents promising empirical results that align well with the theoretical analysis.

**Weaknesses:**

**Methodology**
- Section 4.3 is significantly underdeveloped and requires more extensive elaboration. Expanding on this section with a detailed explanation of the method would improve clarity and depth.
  - In line 279, the paper states, “The proof can be found in ??,” but the proof is missing, and there is no supplementary material to refer to.
  - The proofs for Lemma 4.3 and Lemma 4.4 are also missing. Furthermore, in line 304, "The proof is provided in Appendix ??." mentions that the proof is in the appendix, but no appendix is provided in the paper.
  -  I suggest that the authors include these proofs in an appendix or supplementary material and ensure that all references to proofs are properly linked.

-  The samples that slow down convergence, termed "draggers," seem to refer not to those with gradients perpendicular to the benign samples, but rather those with gradients in the opposite direction (i.e., cosine similarity around -1); in Figure 1, there appear to be cases where benign samples have negative cosine similarity. Are there no samples with reverse gradient directions?

- As noted in the paper's limitations, determining the exact micro-batch size $b$ that achieves the "sweet spot" is a challenging task. This poses a significant empirical limitation, and even within the paper, the optimal micro-batch size changes due to various factors:
  - The best micro-batch size varies by modality; for example, the authors use a size of 4 for the speech task, 32 for the vision task, and 16 for the language task.
  - Additionally, memory constraints may impede the selection of the optimal micro-batch size. For the vision task, a micro-batch size of 32 was used per core, but such size may be difficult to adjust due to memory limitations.
  - I suggest the authors discuss potential strategies or heuristics for selecting micro-batch sizes in practice, considering the observed
variability and memory constraints.

- Along with batch size, one of the most critical hyperparameters in training neural networks is the learning rate. Prior research has explored the relationship between batch size and learning rate [1].

- Is micro-batch clipping independent of mini-batch size, as opposed to micro-batch size? Further investigation into this relationship could be beneficial.

- While this paper analyzes micro-batch clipping in the context of SGD, in practice, SGD is rarely used for training neural networks. It is important to determine whether a similar theoretical analysis holds for modern optimizers like Adam.

- In Section 5.1, to fine-tune the model, the LibriSpeech dataset and handcrafted canaries were synthetically used as benign and dragger samples, respectively. However, this setup seems artificial. In a more realistic scenario where only the LibriSpeech dataset is used, how would dragger samples be detected in practice?  Including practical strategies for detecting draggers would add relevance to the findings.

**Experiments**
- In Tables 2 and 3, experiments were conducted with only batch sizes of 1, 4, and 512, which is insufficient. The authors should have included batch sizes such as 1, 4, 16, 64, 256, and 1024 to provide a more comprehensive evaluation. If these additional results are not feasible, explaining the rationale behind the specific batch sizes chosen would help clarify this decision.

- The analysis related to $|| \mu || \le C || g||$ is completely missing in Section 5.2. I suggest the authors include this analysis in the section or provide an explanation if it was intentionally omitted.

- How does micro-batch clipping perform when training from scratch instead of fine-tuning? This could offer valuable insights into its effectiveness in different training scenarios.

- When conducting the language task experiments, reporting validation performance is inappropriate. It would have been better to test on a separate dataset. Additionally, the paper states that validation set information was not used at all, so how was hyperparameter optimization carried out? Clarifying this would improve transparency.

- The experimental results shown in Table 5 reveal little to no improvement in performance. It would be helpful to discuss possible reasons for this outcome and consider further refinements or additional evaluation metrics.

**Typo**
- The term "nominator" in line 395 should be corrected to "numerator."

- The equation $ C = \sqrt{\frac{2(1-\epsilon)\eta - (1 - \epsilon^2) L \eta^2}{2 \epsilon^2 L \eta^2}}$ in line 219 should be updated to $ C = \sqrt{\frac{2(1-\epsilon)\eta - (1 - \epsilon)^2 L \eta^2}{2 \epsilon^2 L \eta^2}}. $

- Additionally, the expression $ \frac{2(1-\epsilon)\sigma^2}{LB} $ in line 232 should be updated to $ \frac{(1-\epsilon)\sigma^2}{2LB}. $

[1] Huo et al. "Large Batch Optimization for Deep Learning Using New Complete Layer-Wise Adaptive Rate Scaling." AAAI 2021.

**Questions:**

- How are micro-batches grouped within a mini-batch? Is this process conducted randomly?

- Around line 114, the text states that mini-batch gradients are "summed," but isn't it more accurate to describe this as "averaged" or "aggregated," given that the mean is typically computed?

---

> ### Author Response · Authors · 2024-11-20
> **Part 1: thanks and response to G2Xp's review**
>
> We appreciate Reviewer G2Xp's detailed and insightful feedback. We address the reviewer’s specific questions below, and have also corrected the typos identified in the latest version of the paper. We hope our responses have clarified the points the reviewer raised and would be grateful if the reviewer would consider increasing the score.
>
> —---------------------------------------------------------------------------------------------------------------------------------
>
> Q1: Missing appendix
>
> A1: Thank you for catching that! We have added the appendix, including proofs for Lemma 4.3 and 4.4, to the latest version on OpenReview.
>
> —---------------------------------------------------------------------------------------------------------------------------------
>
> Q2: Orthogonal instead of opposite draggers?
>
> A2: This relates to a key observation in our work. Gradients in the opposite direction are less detrimental because adaptive optimizers like Adam adjust learning rates accordingly. However, no current optimizer can fully correct for "dragger" gradients orthogonal to the optimal convergence direction. These drag the model towards irrelevant directions, which is more harmful in the current optimizer landscape. Although our analysis is on SGD for simplicity, this is the intuition behind that drives the choice of orthogonal gradients as draggers.
>
> —---------------------------------------------------------------------------------------------------------------------------------
>
> Q3: Along with batch size, one of the most critical hyperparameters in training neural networks is the learning rate. Prior research has explored the relationship between batch size and learning rate [1]. Is micro-batch clipping independent of mini-batch size, as opposed to micro-batch size? Further investigation into this relationship could be beneficial.
>
> [1] Huo et al. "Large Batch Optimization for Deep Learning Using New Complete Layer-Wise Adaptive Rate Scaling." AAAI 2021.
>
> A3: Thanks for the insightful comment! We’ll add discussion on [1] and this relationship in the revised version.
>
> —---------------------------------------------------------------------------------------------------------------------------------
>
> Q4: In Tables 2 and 3, experiments were conducted with only batch sizes of 1, 4, and 512, which is insufficient. The authors should have included batch sizes such as 1, 4, 16, 64, 256, and 1024 to provide a more comprehensive evaluation. If these additional results are not feasible, explaining the rationale behind the specific batch sizes chosen would help clarify this decision.
>
> A4: We selected these micro-batch sizes for specific reasons. Sizes 1 and 4 fit within a single compute unit, while 512 equals the mini-batch size. These scenarios avoid the complexity of merging gradients from subgroups of compute units, which would require significant low-level changes to our code framework.
>
> —---------------------------------------------------------------------------------------------------------------------------------
>
> Q5: When conducting the language task experiments, reporting validation performance is inappropriate. It would have been better to test on a separate dataset. Additionally, the paper states that validation set information was not used at all, so how was hyperparameter optimization carried out? Clarifying this would improve transparency.
>
> A5: The validation set was not used for training or hyperparameter tuning, ensuring unbiased results. We report validation performance because the SuperGLUE evaluation set is private and requires model submission to their website. Our affiliation's policy prevents us from sharing trained models externally, hence the use of validation set results.
>
> —---------------------------------------------------------------------------------------------------------------------------------
>
> Q6: The experimental results shown in Table 5 reveal little to no improvement in performance. It would be helpful to discuss possible reasons for this outcome and consider further refinements or additional evaluation metrics.
>
> A6: We acknowledged the improvement is marginal in Table 5 and discussed the potential reason in the Evaluation Results section of 5.3.2: “This suggests varying gradient distributions across subsets, with micro-batch clipping suppressing gradients from smaller subsets, which conforms with the observation that ReCoRD has lower accuracy when trained with other data mixed (83.9%) than when trained alone (84.5%)....These results highlight a crucial lesson: micro-batch clipping, while beneficial when data comes from the same domain but is noisy, can be detrimental with unbalanced multi-domain data. In such scenarios, micro-batch clipping may hinder certain domains to favor others and should be avoided.”, and we verified the hypothesis in Table 6.

---

> > ### Author Response · Authors · 2024-11-20
> > **Part 2: thanks and response to G2Xp's review**
> >
> > Q7: How are micro-batches grouped within a mini-batch? Is this process conducted randomly?
> >
> > A7: Yes, the grouping is random. We will clarify this in the revised version.
> >
> > —---------------------------------------------------------------------------------------------------------------------------------
> >
> > Q8: Around line 114, the text states that mini-batch gradients are "summed," but isn't it more accurate to describe this as "averaged" or "aggregated," given that the mean is typically computed?
> >
> > A8: This is to be consistent with our Algorithm 1 and Algorithm 1 in our main reference [1]. Either summing or averaging does not affect the asymptotic behavior of convergence.
> >
> > [1] Automatic Clipping: Differentially Private Deep Learning Made Easier and Stronger.
> >
> > —---------------------------------------------------------------------------------------------------------------------------------
> >
> > Q9: The analysis related to ||μ||≤C||g|| is completely missing in Section 5.2. I suggest the authors include this analysis in the section or provide an explanation if it was intentionally omitted.
> >
> > A9: We omitted the analysis for C because it doesn't affect our key asymptotic conclusions and is computationally expensive, requiring logging all gradients at each step. We focused on key parameters like c, which influences our explanation of the "sweet spot" for micro-batch size.
> >
> > —---------------------------------------------------------------------------------------------------------------------------------
> >
> > Q10: Add discussion potential strategies or heuristics for selecting micro-batch sizes in practice
> >
> > A10: Thanks for the insightful feedback. We’ll add the discussion in the revised version.
> >
> > —---------------------------------------------------------------------------------------------------------------------------------
> >
> > Q11: While this paper analyzes micro-batch clipping in the context of SGD, in practice, SGD is rarely used for training neural networks. It is important to determine whether a similar theoretical analysis holds for modern optimizers like Adam.
> >
> > A11: We appreciate this feedback and will address it in the future work section of the revised version.
> >
> > —---------------------------------------------------------------------------------------------------------------------------------
> >
> > Q12: In Section 5.1, to fine-tune the model, the LibriSpeech dataset and handcrafted canaries were synthetically used as benign and dragger samples, respectively. However, this setup seems artificial. In a more realistic scenario where only the LibriSpeech dataset is used, how would dragger samples be detected in practice? Including practical strategies for detecting draggers would add relevance to the findings.
> >
> > A12: Thanks for the insightful feedback! The reason we use handcrafted canaries is that it’s hard to identify natural draggers because the draggers might change across iterations (for example, some examples are draggers in the early stage of training, and some become draggers in the later stage of training). Besides, some natural triggers might not be perfectly orthogonal but a mixture of gradients in the 2 directions. To simplify the experiment setup, we choose to use extreme outliers to show strong signals.
> >
> > —---------------------------------------------------------------------------------------------------------------------------------
> >
> > Q13: How does micro-batch clipping perform when training from scratch instead of fine-tuning? This could offer valuable insights into its effectiveness in different training scenarios.
> >
> > A13: Training models from scratch is too computationally expensive. We will consider this for future work if our budget allows.

---

> ### Comment · Reviewer_G2Xp · 2024-11-26
> **Thank you for your detailed responses**
>
> Thank you for your detailed feedback. Most of my concerns have been addressed. I would consider increasing the score if you could provide more detailed responses to the points below without necessarily conducting additional experiments. Additionally, the manuscript would benefit from overall proofreading and revision.
>
> - I suggest the authors discuss potential strategies or heuristics for selecting micro-batch sizes in practice, considering the observed variability and memory constraints.
>
> - Along with batch size, one of the most critical hyperparameters in training neural networks is the learning rate. Prior research has explored the relationship between batch size and learning rate [1]. Is micro-batch clipping independent of mini-batch size, as opposed to micro-batch size? Further investigation into this relationship could be beneficial.
>
> - In Tables 2 and 3, experiments were conducted with only batch sizes of 1, 4, and 512, which is insufficient. The authors should have included batch sizes such as 1, 4, 16, 64, 256, and 1024 to provide a more comprehensive evaluation. If these additional results are not feasible, explaining the rationale behind the specific batch sizes chosen would help clarify this decision.

---

> ### Author Response · Authors · 2024-11-26
> **Part 1: thanks for the follow-up questions**
>
> Thank you for taking the time to review the rebuttal! We're happy to provide more detailed responses to the questions we do not thoroughly answered in the first rebuttal. We'll also be updating the PDF with a concise version of these answers soon.
>
> ---------------------------------------------------------------------------------------------------------------------------
>
> Q1: Potential strategies or heuristics for selecting micro-batch sizes in practice.
>
> A1: We appreciate the reviewer's suggestion. The variability in micro-batch size selection is indeed a crucial aspect that we should address more thoroughly in the paper.
>
> In practice, our heuristics for choosing the micro-batch size is mainly dependent on three metrics, memory, compute rounds and performance.
>
> - Memory wise, choosing a batch size that's a multiple of the per-core batch size doesn't increase memory usage for micro-batches. Any size smaller than the per-core batch size will potentially increase the memory overhead, leading to the need for more compute units (GPUs/TPUs).
>
> - Compute-round wise, choosing a batch size smaller than or equal to the per-core batch size maintains all the computation locally without adding additional synchronization rounds. Choosing anything larger than the per-core batch size will add an additional rounds of synchronization to conduct averaging and clipping, and also potentially more coding as synchronization across not all but a few compute units are not supported in most of the deep learning libraries off-the-shelf.
>
> The above 2 constraints, although not important in convergence analysis, are extremely important in practice because they affect the need for expensive compute resources and training time. The only choice that does not increase either is to directly use the default per-core batch size as micro-batch size, which is also what we use in all the experiments in the paper except the ablation studies on micro-batch size.
>
> - Performance-wise, surprisingly, using the per-core batch size as micro-batch size always give positive results in all our three experiments. This might indicate that although a sweet spot exists, the micro-batch size that can bring improvements are not rare. If compute is a more strict constraint and the original per-core batch size lies in a reasonable range (not 1 and also not too large to be close to the mini-batch size), then using per-core batch size will very likely improve the performance without any compute overhead.
>
> ---------------------------------------------------------------------------------------------------------------------------
> Q2: Prior research has explored the relationship between batch size and learning rate [1]. Is micro-batch clipping independent of mini-batch size, as opposed to micro-batch size?
>
> [1] Huo et al. "Large Batch Optimization for Deep Learning Using New Complete Layer-Wise Adaptive Rate Scaling." AAAI 2021.
>
> A2: That's an insightful question. The interaction between learning rate and micro-batch size (and also clipping bound) is indeed an interesting avenue for future exploration. In [1], the authors found that some learning rate tricks such as warmup and CLARS can be explained by alleviating the training difficulties caused by gradient variance. In our setting with micro-batch clipping, the gradient variance is now not only affected by mini-batch size, but also another two hyper-parameters: micro-batch size and clipping bound. Although it’s out of the paper’s scope to develop a complete theory for this relationship, some of our educated hypotheses are:
>
> - Learning rate should remain related to mini-batch size when micro-batch size is fixed, because no matter what distribution the clipped micro-batch gradient is, increasing the number of micro-batches will always shrink the gradient variance according to the law of large numbers.
> - Learning rate should be related to micro-batch size in the same asymptotic trend as mini-batch size, because the larger the micro-batch size, the more concentrated the clipped micro-batch gradients will be and the smaller the final gradient variance will be.
> - Learning rate’s relationship to clipping bound seems more complex as whether the clipping will decrease or increase gradient variance depends on the distribution of the gradients. Let’s use two gradients as a simple example. If one gradient is large and the other one is small but in the opposite direction, clipping with tighter bounds will increase the gradient variance/clipping bound ratio until both gradients are clipped, from where no progress will be made because the average gradient is always 0. On the other hand, if the two gradients are in the same direction, then clipping them with tighter bound will decrease the gradient variance/clipping bound ratio.

---

> > ### Author Response · Authors · 2024-11-26
> > **Part 2: thanks for the follow-up questions**
> >
> > Q3: In Tables 2 and 3, experiments were conducted with only batch sizes of 1, 4, and 512, which is insufficient. The authors should have included batch sizes such as 1, 4, 16, 64, 256, and 1024 to provide a more comprehensive evaluation. If these additional results are not feasible, explaining the rationale behind the specific batch sizes chosen would help clarify this decision.
> >
> > A3: We acknowledge that exploring a wider range of micro-batch sizes would provide a more complete picture of micro-batch clipping behavior. Our initial selection of 1, 4, and 512 was driven by a desire to simplify the experimental setup and focus on the core principles of micro-batch clipping.
> > Specifically, these sizes allowed us to:
> > - Establish a baseline: Using a micro-batch size equaling mini-batch size (almost equivalent to standard SGD) provides a clear baseline to assess the impact of micro-batch clipping.
> > - Align with hardware constraints: A micro-batch size of 4 aligns with the per-core batch size in our hardware setup, enabling efficient computation within individual compute units without any memory overhead/additional synchronization rounds. Like we mentioned in the response to Q1, using micro-batch 16, 64 and 256 would require an additional synchronization round which slows down the training.
> > - Examine the border: Setting the micro-batch size to 1 allows us to observe the behavior under the lower bound of micro-batch size.
> >
> > While these choices facilitated our initial investigation, we recognize the value in examining intermediate micro-batch sizes. Introducing sizes like 16, 64, and 256 could reveal more nuanced performance trends and interactions with the mini-batch size. However, incorporating these would necessitate significant modifications to our codebase to handle gradient merging across compute units, a complexity we aimed to avoid in this initial exploration.
> > We believe our current experiments provide valuable insights into the core mechanisms of micro-batch clipping. Future work could expand on this by addressing the complexities of a wider range of micro-batch sizes and their interaction with the underlying hardware and software infrastructure.

---

> > > ### Comment · Reviewer_G2Xp · 2024-11-27
> > > **Thank you again for your responses**
> > >
> > > I appreciate your comprehensive response, which has successfully addressed the majority of my concerns. Based on the thoroughness of your revisions, I have decided to adjust my review score upward from 5 to 6.

---

> > > > ### Author Response · Authors · 2024-11-27
> > > > **thanks for the discussion!**
> > > >
> > > > Thank you very much for your prompt and insightful response. Your feedback has been invaluable in improving the manuscript.

---

### Official Review · Reviewer_nmLX · 2024-11-01

**Soundness:** 3
**Presentation:** 3
**Contribution:** 3
**Rating:** 6
**Confidence:** 3

**Summary:**

Micro-batch clipping has recently demonstrated potential in improving model performance in automatic speech recognition (ASR). This paper seeks to uncover the mechanism driving this enhancement. The authors assume that certain data samples ("draggers") may impede model convergence. Through theoretical analysis, they find that micro-batch clipping can accelerate convergence by diminishing the impact of these draggers, albeit introducing a constant bias dependent on micro-batch size. This insight further explains the observed phenomenon where improvements occur only at specific micro-batch sizes. Additional empirical experiments are conducted beyond ASR to investigate the effectiveness of micro-batch clipping.

**Strengths:**

- Uncovering the mechanism behind the performance improvements brought by micro-batch clipping is both interesting and valuable.
- The paper provides a series of theoretical foundation to analyze this mechanism and support its conclusions.
- The theoretical findings in this paper can explain empirical results, such as the performance gains achieved through micro-batch clipping and why these improvements occur only at specific micro-batch sizes.
- Extensive experiments are conducted to explore the benefits of micro-batch clipping beyond ASR.

**Weaknesses:**

- The Appendix of this paper is somewhat lost, and the full proofs of the theorems are thus invisible, requiring further verification.
- Assumption 4.4 posits that dragger gradients are orthogonal to the benign gradient subspace, which may be a restrictive assumption and not entirely realistic in practical scenarios.
- In Sec.1, the authors claim that micro-batch clipping can adaptively suppress samples that hinder convergence. However, it remains somewhat unclear in the theorem whether the effectiveness of micro-batch clipping directly stems from suppressing these draggers.
- It appears that as $\epsilon$ decreases, approaching zero and indicating a lower likelihood of draggers, the convergence rate bound in Thm 4.2 actually increases, which seems counterintuitive.

**Questions:**

Please refer to the weakness section.

---

> ### Author Response · Authors · 2024-11-20
> **thanks and reponse to nmLX's review**
>
> We’d like to thank the reviewers for the positive feedback and insightful suggestions.
>
> —---------------------------------------------------------------------------------------------------------------------------------
>
> Q1: Missing appendix.
>
> A1: Thanks for the catch! we’ve added back the appendix in the latest version. Please check.
>
> —---------------------------------------------------------------------------------------------------------------------------------
>
> Q2: Assumption 4.4 posits that dragger gradients are orthogonal to the benign gradient subspace, which may be a restrictive assumption and not entirely realistic in practical scenarios.
>
> A2: We appreciate your insightful feedback. We acknowledge that the orthogonality assumption simplifies real-world scenarios. However, this simplification enables analysis that would be significantly more complex or even infeasible otherwise. Similar assumptions, like Assumption 5.3 on symmetry, are found in previous works. Furthermore, this assumption is empirically supported by the results shown in Figure 1. We will explicitly mention this limitation in the revised version.
>
> —---------------------------------------------------------------------------------------------------------------------------------
>
> Q3: In Sec.1, the authors claim that micro-batch clipping can adaptively suppress samples that hinder convergence. However, it remains somewhat unclear in the theorem whether the effectiveness of micro-batch clipping directly stems from suppressing these draggers.
>
> A3: Thanks for the insightful comment! The effectiveness comes from the clipping operation which unifies the size of gradients, especially the draggers gradients which are usually larger. Besides, the explanation of the existence of the sweet point hinges on the existence of the dragger terms.
>
> —---------------------------------------------------------------------------------------------------------------------------------
>
> Q4: It appears that as ϵ decreases, approaching zero and indicating a lower likelihood of draggers, the convergence rate bound in Thm 4.2 actually increases, which seems counterintuitive.
>
> A4: In Theorem 4.2, we operate under the assumption that $\epsilon\geq\frac{1}{b+1}$​. Our analysis is therefore applicable within that specific range.

---

> > ### Comment · Reviewer_nmLX · 2024-11-26
> >
> > I appreciate the authors' efforts in addressing the concerns raised in the rebuttal. My questions have been resolved. I decide to maintain my rating.

---

### Official Review · Reviewer_stJ2 · 2024-11-02

**Soundness:** 3
**Presentation:** 3
**Contribution:** 3
**Rating:** 6
**Confidence:** 4

**Summary:**

This paper revisits micro-batch clipping, an optimization technique initially proposed for memory efficiency in differentially private stochastic gradient descent. The paper conceptualizes micro-batch clipping as a form of data pruning, proposing that certain "dragger" gradients impede model convergence. Through convergence analysis and empirical evaluation, the paper demonstrates that micro-batch clipping can asymptotically accelerate convergence but introduces a constant bias term. This bias term explains the existence of an optimal micro-batch size. The paper further validates the effectiveness of micro-batch clipping beyond speech models, showing promising results on vision and language tasks. However, limitations are identified when training data originates from multiple distinct domains.

**Strengths:**

1. Novel Conceptualization: The paper innovatively frames micro-batch clipping as a data pruning technique, introducing the concept of "dragger" gradients to explain its mechanism. This provides new insights into why and when micro-batch clipping works.
2. Comprehensive Analysis: The paper conducts both theoretical convergence analysis and extensive empirical evaluation to support its hypotheses. The results demonstrate the effectiveness of micro-batch clipping across different model architectures and datasets.

**Weaknesses:**

1. Lack of interpretability or robustness. This paper does not discuss the potential impact of micro-batch clipping on the interpretability or robustness of the trained models, which are important considerations in many real-world applications.
2. Limited scope of experiments: The experiments are primarily conducted on speech recognition models and limited vision and language tasks. The generalizability of the findings to a broader range of models and datasets is unclear. More extensive experimentation across different domains and model architectures would strengthen the conclusions.
3. Concern of memory overhead: While the paper mentions that data sharding with a micro-batch size equal to the per-core batch size can mitigate memory overhead, this may not be feasible for larger models. The growing size of modern models necessitates alternative solutions to ensure computational efficiency.
4. Rigorous relationship between micro-batch size and performance: The paper suggests that there exists an optimal micro-batch size that minimizes the constant bias term in the convergence analysis. However, the relationship between the micro-batch size b and the factor c is not rigorously established. A tighter convergence bound and a more accurate relationship between b and c could better guide the choice of micro-batch sizes in practice.

**Questions:**

See Weaknesses.

---

> ### Author Response · Authors · 2024-11-20
> **thanks and response to stJ2's review**
>
> We’d like to thank the reviewer for the positive feedback and insightful suggestions.
>
> —---------------------------------------------------------------------------------------------------------------------------------
>
> Q1: Lack of interpretability or robustness.
>
> A1: We appreciate the reviewer's insight. While our primary focus was on convergence rates, we acknowledge the importance of robustness and interpretability and will add these in the future work section.
>
> —---------------------------------------------------------------------------------------------------------------------------------
>
> Q2: Limited scope of experiments.
>
> A2: Thank you for this observation. Some experiments, such as Figure 1, require domain-specific adaptations (e.g., the use of canaries). Our findings on micro-batch clipping accelerating convergence are also rooted in the speech domain, hence our focus there. Besides, most of the experiments in the paper are computationally expensive to run so it’s beyond our budget to add similar-scale experiments. We acknowledge the value of broader experiments and will add this in the future work section.
>
> —---------------------------------------------------------------------------------------------------------------------------------
>
> Q3: Concern of memory overhead for larger models.
>
> A3: Thank you for raising this point. To clarify, when the micro-batch size matches the per-core batch size, there's virtually no memory overhead, regardless of model size. For instance, when the model is large enough such that the per-core batch size is 1, using a micro-batch size of 1 doesn't increase memory. If the model is too large that the model has to be sharded across a few compute cores, it does introduce an extra synchronization round for gradient norm calculation, which is a computational rather than memory overhead. We'll add the explanation in the revised version.
>
> —---------------------------------------------------------------------------------------------------------------------------------
>
> Q4: Rigorous relationship between micro-batch size and performance.
>
> A4: We appreciate the reviewer's insight. We acknowledge the need to further investigate the precise relationship between micro-batch size and performance and will address this as a limitation and future research direction. While we explored establishing an explicit relationship, a rigorous analysis proved challenging within our current mathematical framework without incurring new assumptions.

---

### Author Response · Authors · 2024-11-24
**A kind reminder on response to rebuttal**

We'd like to thank you again for your diligent work in reviewing our manuscript and providing valuable feedback.

As the discussion deadline approaches, we kindly request that you take another look at our response. If our response has addressed your initial concerns, we would be grateful if you would consider adjusting your score accordingly.

We are particularly grateful that you pointed out the missing appendix issue. We have now fixed it in the latest version of the manuscript on OpenReview.

---

### Meta-Review · Area_Chair_pJMV · 2024-12-21

**Metareview:**

This paper revisits micro-batch clipping, an optimization technique initially introduced for improving memory efficiency in differentially private stochastic gradient descent. The authors conceptualize micro-batch clipping as a form of data pruning, proposing that certain "dragger" gradients hinder model convergence. Through both convergence analysis and empirical evaluation, the paper demonstrates that micro-batch clipping can asymptotically accelerate convergence, although it introduces a constant bias term. This bias term helps explain the existence of an optimal micro-batch size.

Additionally, the paper validates the effectiveness of micro-batch clipping beyond speech models, showing promising results on vision and language tasks. However, the authors identify limitations when training data comes from multiple distinct domains. The paper offers an innovative perspective by framing micro-batch clipping as a data pruning technique and introducing the concept of "dragger" gradients to explain the underlying mechanism. This approach provides valuable insights into why and when micro-batch clipping is effective.

However, Assumption 4.4, which posits that dragger gradients are orthogonal to the benign gradient subspace, may be a restrictive assumption that doesn't fully reflect practical scenarios. Furthermore, it appears that as the dragger gradient parameter decreases, approaching zero and suggesting a lower likelihood of draggers, the convergence rate bound in Theorem 4.2 actually increases. This result seems counterintuitive, raising questions about the assumptions and implications of the analysis.

**Additional Comments On Reviewer Discussion:**

Reviewers raised several questions about the paper, including:

- **Lack of interpretability and robustness**: The potential impact of micro-batch clipping on the interpretability or robustness of trained models has not been thoroughly examined, leaving an important aspect of its practical implications unexplored.
- **Limited experimental scope**: While the rebuttal addressed some concerns, the experiments remain primarily focused on speech recognition models, with limited exploration of vision and language tasks, raising questions about generalizability.
- **Unclear relationship between micro-batch size and performance**: Although the paper identifies an optimal micro-batch size to minimize bias, the relationship between micro-batch size and performance lacks rigorous analysis.

After the rebuttal, the authors successfully addressed these questions to the satisfaction of all reviewers. However, some suggestions were made to further enhance the paper:

1. **Discussion on micro-batch size selection**: The authors are encouraged to discuss potential strategies or heuristics for selecting optimal micro-batch sizes in practice, taking into account variability in performance and memory constraints.

2. **Exploration of batch size and learning rate relationships**: Since batch size and learning rate are critical hyperparameters in neural network training, the paper could benefit from investigating whether micro-batch clipping depends on mini-batch size or operates independently. Prior research on the relationship between batch size and learning rate [1] could provide a foundation for such an exploration, offering deeper insights and practical guidance.

---

### Decision · Program_Chairs · 2025-01-22

Accept (Poster)